# *Crocosphaera* as a Major Consumer of Fixed Nitrogen

Takako Masuda,[a,b]* Keisuke Inomura,[c] Taketoshi Kodama,[a] Takuhei Shiozaki,[a]§ Satoshi Kitajima,[a]◊ Gabrielle Armin,[c] Takato Matsui,[d] Koji Suzuki,[d] Shigenobu Takeda,[a]∞ Mitsuhide Sato,[a]‡ Ondřej Prášil,[b] Ken Furuya[a]#

[a]Department of Aquatic Bioscience, The University of Tokyo, Tokyo, Japan
[b]Institute of Microbiology, The Czech Academy of Sciences, Třeboň, Czech Republic
[c]Graduate School of Oceanography, University of Rhode Island, Narragansett, Rhode Island, USA
[d]Graduate School of Environmental Science/Faculty of Environmental Earth Science, Hokkaido University, Sapporo, Japan

Takako Masuda and Keisuke Inomura contributed equally to the article. The author order was determined based on the chronology of the project.

**ABSTRACT** *Crocosphaera watsonii* (hereafter referred to as *Crocosphaera*) is a key nitrogen (N) fixer in the ocean, but its ability to consume combined-N sources is still unclear. Using *in situ* microcosm incubations with an ecological model, we show that *Crocosphaera* has high competitive capability both under low and moderately high combined-N concentrations. In field incubations, *Crocosphaera* accounted for the highest consumption of ammonium and nitrate, followed by picoeukaryotes. The model analysis shows that cells have a high ammonium uptake rate ($\sim$7 mol N [mol N]$^{-1}$ d$^{-1}$ at the maximum), which allows them to compete against picoeukaryotes and nondiazotrophic cyanobacteria when combined N is sufficiently available. Even when combined N is depleted, their capability of nitrogen fixation allows higher growth rates compared to potential competitors. These results suggest the high fitness of *Crocosphaera* in combined-N limiting, oligotrophic oceans heightening its potential significance in its ecosystem and in biogeochemical cycling.

**IMPORTANCE** *Crocosphaera watsonii* is as a key nitrogen (N) supplier in marine ecosystems, and it has been estimated to contribute up to half of oceanic N$_2$ fixation. Conversely, a recent study reported that *Crocosphaera* can assimilate combined N and proposed that unicellular diazotrophs can be competitors with non-N$_2$ fixing phytoplankton for combined N. Despite its importance in nitrogen cycling, the methods by which *Crocosphaera* compete are not currently fully understood. Here, we present a new role of *Crocosphaera* as a combined-N consumer: a competitor against nondiazotrophic phytoplankton for combined N. In this study, we combined *in situ* microcosm experiments and an ecosystem model to quantitatively evaluate the combined-N consumption by *Crocosphaera* and other non-N$_2$ fixing phytoplankton. Our results suggest the high fitness of *Crocosphaera* in combined-N limiting, oligotrophic oceans and, thus, heightens its potential significance in its ecosystem and in biogeochemical cycling.

**KEYWORDS** *Crocosphaera watsonii*, marine N$_2$ fixer, combined nitrogen, ecological model

Marine phytoplankton contribute about one-half of the global net primary production and play a key role in regulating global biogeochemical cycles (1). Since phytoplankton are biochemically, metabolically, and ecologically diverse (2–4), understanding the contribution of different phytoplankton groups to ecosystem function is central to the precise estimation of the global carbon (C) and nitrogen (N) budget and to predicting the biogeochemical impact of future environmental changes (5).

In the oligotrophic subtropical gyres, combined N (defined as N covalently bonded to one or more elements other than N [6]) limits primary production and controls planktonic community composition (7–10). Therefore, N$_2$-fixing microorganisms (diazotrophs) are important as a source of combined N in oligotrophic ecosystems (11, 12). In the subtropic

Address correspondence to Takako Masuda, takakom@affrc.go.jp.

*Present address: Takako Masuda, Fisheries Resources Institute, Japan Fisheries Research and Education Agency, Shiogama, Miyagi, Japan.

§Present address: Takuhei Shiozaki, Atmosphere and Ocean Research Institute, The University of Tokyo, Kashiwa, Chiba, Japan.

◊Present address: Satoshi Kitajima, Fisheries Resources Institute, Japan Fisheries Research and Education Agency, Nagasaki, Japan.

∞Present address: Shigenobu Takeda, Graduate School of Fisheries and Environmental Sciences, Nagasaki University, Nagasaki, Japan.

‡Present address: Mitsuhide Sato, Graduate School of Fisheries and Environmental Sciences, Nagasaki University, Nagasaki, Japan.

#Present address: Ken Furuya, Graduate School of Science and Engineering, Soka University, Tokyo, Japan.

The authors declare no conflict of interest.

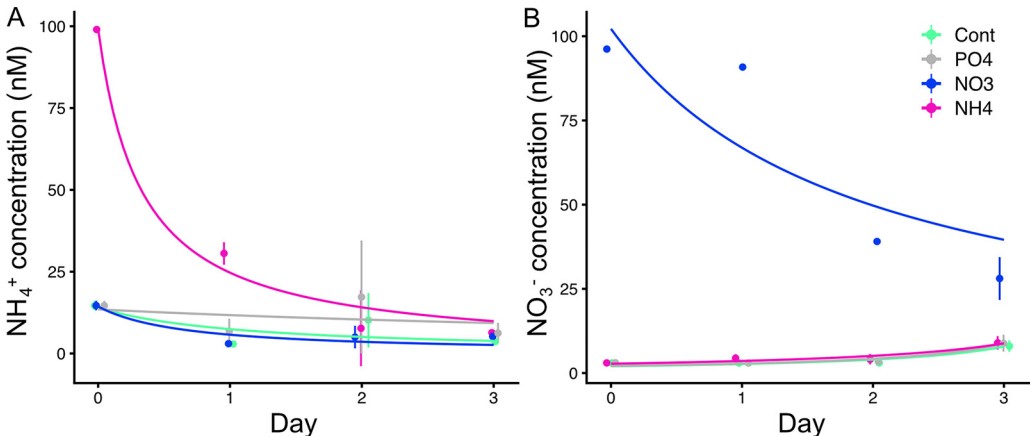

**FIG 1** Temporal change in NH$_4^+$ and NO$_3^-$ concentrations of experiment M3. (A) NH$_4^+$ concentration in the NH$_4^+$ treatment exponentially decreased during the experiment down to the detection limit of 6 nM on day 3. (B) NO$_3^-$ concentrations in the NO$_3^-$ treatment exponentially decreased during the experiment, but enriched NO$_3^-$ was not always entirely consumed. Error bar shows a standard deviation of triplicate. Temporal change in urea-N concentration is shown in Fig. S2 in the supplemental material.

oligotrophic ocean, the unicellular diazotroph, *Crocosphaera watsonii* (2.5 to 6 $\mu$m; hereafter referred to as *Crocosphaera*), is widely distributed (10, 13–16) in addition to pico-sized (<3 $\mu$m) cyanobacteria (e.g., *Prochlorococcus* and *Synechococcus*) and picoeukaryotes (17–19). Earlier studies examined the effect of combined N, such as ammonium (NH$_4^+$) and nitrate (NO$_3^-$), on metabolic activities and reveal the ability of *Crocosphaera* to assimilate combined N (20, 21). As reported from *Trichodesmium* (22), increasing concentrations of NH$_4^+$ enrichment increases NH$_4^+$ uptake activities and inhibits N$_2$ fixation rates up to ~80% (20, 21), while NO$_3^-$ enrichment did not inhibit N$_2$ fixation rate at any of the tested NO$_3^-$ concentrations (up to 10 $\mu$M) (20). When remaining combined-N concentrations in the cultures are at a nanomolar level, *Crocosphaera* kept fixing N$_2$ (20, 21). Model results indicate that using dissolved inorganic nitrogen (DIN) enables *Crocosphaera* populations to increase their abundance and expand their niche (23). These studies proposed that unicellular diazotrophs can be competitors with nondiazotrophic phytoplankton for combined N. However, how *Crocosphaera* competes for combined N is poorly evaluated. In this study, we combine an *in situ* microcosm experiment with N addition at the nanomolar level and model (24) to evaluate the competitiveness of *Crocosphaera* in an N-limiting environment.

## RESULTS

**Summary of the experiment.** We carried out five nitrogen (N) and phosphorus (P) addition bioassays (M1 to M5) every 4 days at a station in the subtropical Northwestern Pacific (12°N, 135°E) from 6 to 25 June 2008 during the MR08-02 cruise on the R/V *MIRAI*. The northward current was dominant until bioassay M3, while a strong southward current occurred on days between bioassays M3 and M4. The initial waters were more oligotrophic during M1 to M3 compared to those during M4 and M5; nutrient concentrations initially were less than 36 nM for ammonium (NH$_4^+$), 7 nM for nitrate plus nitrite (NO$_3^-$ + NO$_2^-$), and 64 nM for phosphorus (PO$_4^{3-}$) (see Table S1 in the supplemental material). The lower initial phytoplankton abundance during M1 to M3 than that during M4 and M5 confirms the oligotrophic characteristics of initial water during M1 to M3 (Table S1). Although we performed prefiltration with a 1-$\mu$m polypropylene cartridge filter (Micropore EU; ORGANO) to eliminate the effect of grazing, water samples contained plankton up to ~5 $\mu$m in size.

**Nutrient uptake and fate of enriched DIN.** For 3 days of incubation, the phytoplankton community consumed NH$_4^+$ entirely at the end, while NO$_3^-$ was not always consumed completely (Fig. 1; see also Fig. S1 in the supplemental material). Estimated biomass explains about half of consumed combined-N sources (Fig. 1, 2A).

The greatest portion of estimated C and N in biomass was found in *Crocosphaera* (39 to 93% in all N addition incubations) followed by picoeukaryotes (5 to 55% in N addition

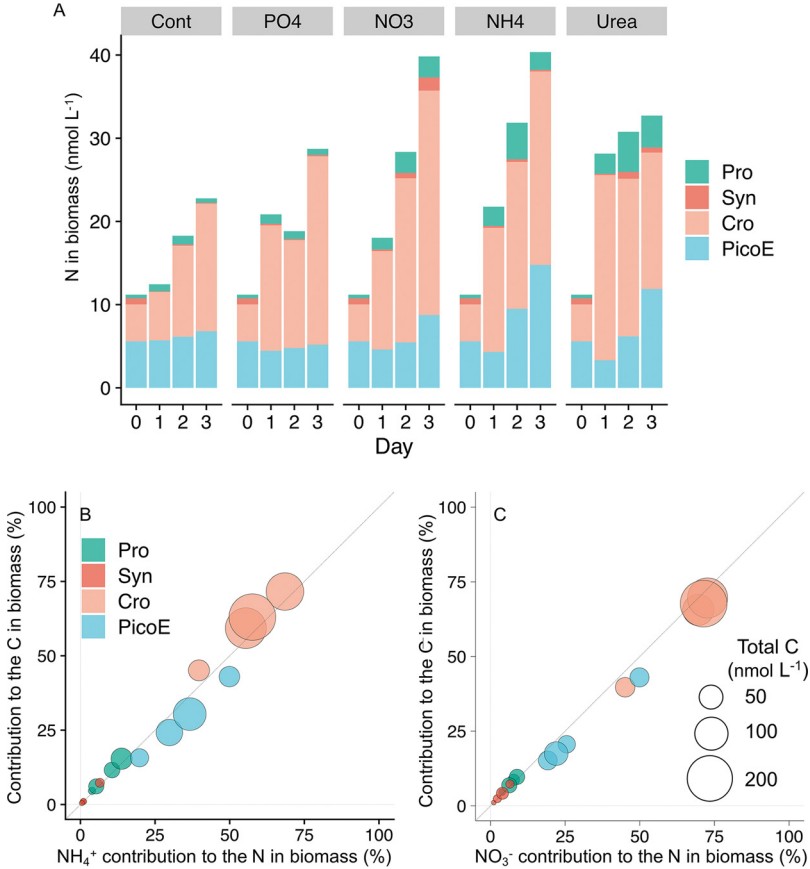

**FIG 2** (A) N in biomass in each treatment and its contribution of each phytoplankton group of experiment M3. (B) Contribution to total C in biomass as a function of the contribution of NH$_4^+$-N biomass for each phytoplankton group. (C) Contribution to total carbon in biomass as a function of the contribution of NO$_3^-$-N biomass for each phytoplankton group. The contributions of NH$_4^+$-N or NO$_3^-$-N were estimated from either NH$_4^+$ or NO$_3^-$ enrichment. Each circle shows data from a different day, and the size of the dots represents the total C in biomass (nmol C L$^{-1}$). Pro, *Prochlorococcus*; Syn, *Synechococcus*; Cro, *Crocosphaera*; PicoE, picoeukaryotes.

incubations) (Fig. 2; see also Fig. S3 and S4 in the supplemental material). Although the origin of water mass changed from oligotrophic water to mixed water between experiments M1 to M3 and M4 to M5 (25), with more *Crocosphaera* in cell density with higher N$_2$ fixation in the latter environment (see Tables S1 and S2 in the supplemental material), the dominance of *Crocosphaera* as a C and N biomass was observed from all of the experiments. N derived from N$_2$ fixation was not always sufficient to support the N demand of *Crocosphaera*, especially in N amendment (see Fig. S5 in the supplemental material). Estimated N$_2$ fixation supported 0.5 to 12.7% of N demand of *Crocosphaera* in control and 0.5 to 11.6% in NH$_4^+$ treatment (Fig. S5), suggesting that *Crocosphaera* consumed amended N sources. Assimilation of combined nitrogen (NH$_4^+$ and NO$_3^-$), together with N$_2$ fixation by *Crocosphaera*, has been reported (20, 21). Although enriched 100 nM NH$_4^+$ was completely consumed (<6 nM; detection limit on day 3), increases in N biomass of nondiazotrophs for 3 days were limited to up to 58 nmol L$^{-1}$, again suggesting *Crocosphaera* took up combined nitrogen.

**Model analysis of the data.** To quantitatively interpret the observed data, we used a simple model of cellular growth, which is based on the uptake of NH$_4^+$ and NO$_3^-$ (see Materials and Methods). It is natural that such *in situ* incubation experiments display a variation in their results, since the initial conditions vary based on the locations. The ideal conditions to test relaxation from nutrient stress are to use a nutrient-starved phytoplankton community, which spends a long time under low-nutrient conditions (26, 27). Considering the nutrient history of the *in situ* phytoplankton community, we selected M3 as the best example to observe relief from nutrient stress, since the water

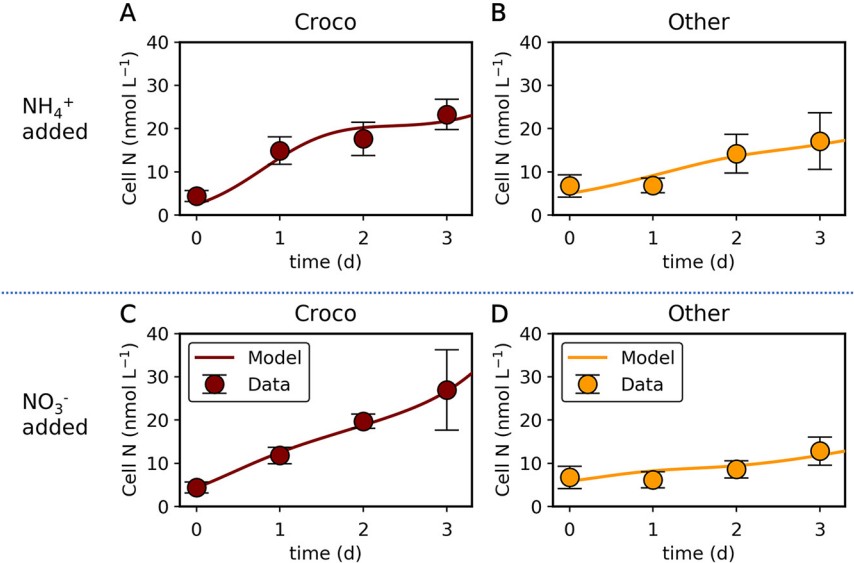

**FIG 3** Simulated transition of cellular N with nutrient addition compared with data. (A, B) NH$_4^+$-added case. (C, D) NO$_3^-$-added case. Croco, *Crocosphaera*; Other, other phytoplankton. Data are from experiment M3.

mass changed from more N-limited to N-rich water between M3 and M4 (see Tables S1 and S3 in the supplemental material). The N-limited nutrient history of phytoplankton was confirmed by low initial nutrient conditions and low biomass of the targeted organisms. Thus, we focus on the data from experiment M3 for modeling analysis.

The model captured the overall trend of the transition of cellular N (Fig. 3) based on the available nutrient (see Fig. S6 in the supplemental material). The parameterization of the model reveals high rates of N uptake by *Crocosphaera*. We used 6.6 mol N (mol N)$^{-1}$ d$^{-1}$ for maximum NH$_4^+$ uptake under NH$_4^+$ limitation to represent the data, which represent high combined-N uptake compared to that of other phytoplankton under the same condition (maximum NH$_4^+$ uptake of 1.1 mol N [mol N]$^{-1}$ d$^{-1}$) (see Table S4 in the supplemental material). Specifically, such parameterization was needed to reproduce the rapid growth of *Crocosphaera* under NH$_4^+$ added conditions between day 0 and day 1. The predicted maximum NO$_3^-$ uptake rate for *Crocosphaera* is also higher than for other phytoplankton (Fig. 3), which is supported by the faster growth of *Crocosphaera* with NO$_3^-$ addition.

To test the competitiveness of *Crocosphaera*, we simulated a simple ecological situation. Here, we simulated zooplankton with kill the winner (KTW) theory (28). We used this method because it reflects the commonly observed active prey-switching behavior of zooplankton (29–31). The result shows the high competitiveness of *Crocosphaera* under both high and low nutrient concentrations (Fig. 4; see also Fig. S7 in the supplemental material). Under a high nutrient concentration, *Crocosphaera* may dominate other phytoplankton due to the high rate of nutrient uptake (Fig. 4A; see also Fig. S7A). However, under extremely low-nutrient conditions (NH$_4^+$ and

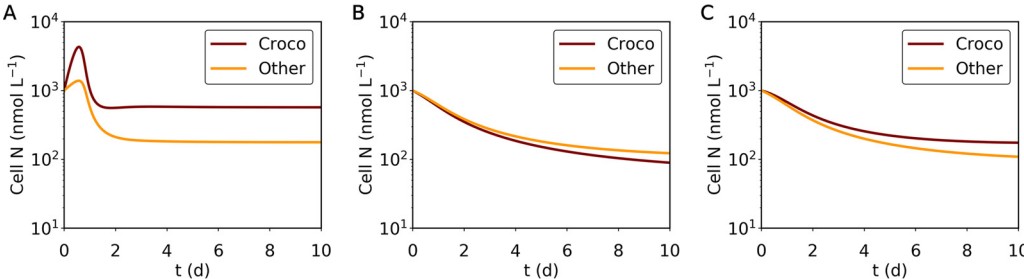

**FIG 4** Simulated transition of cellular N in a simple ecosystem model for three different scenarios. (A) The concentrations for NH$_4^+$ and NO$_3^-$ are both 100 nmol L$^{-1}$. (B, C) The concentrations for NH$_4^+$ and NO$_3^-$ are both 1 nmol L$^{-1}$. In only panel C, *Crocosphaera* may acquire N via N$_2$ fixation; in panels A and B, the effect of N$_2$ fixation is neglected. Parameters are based on the NH$_4^+$-added case.

NO$_3^-$ are both at 1 nmol L$^{-1}$), *Crocosphaera* is slightly outcompeted (Fig. 4B; see also Fig. S7C and D). This is due to the relatively high half-saturation constant for NH$_4^+$, which is manifested by the sudden decrease in growth rate with a drop in NH$_4^+$ under NH$_4^+$ addition (Fig. 3A; see also Fig. S6A). However, this relationship flips if we consider the effect of N$_2$ fixation, which maintains growth rates at a higher level rather than relying on external N under N depletion (Fig. 4C; see also Fig. S7B). These results suggest that possession of nitrogenase (an enzyme complex involved in N$_2$ fixation) allows for the survival of *Crocosphaera* under low-nutrient environments.

## DISCUSSION

Our study shows high uptake of N by *Crocosphaera* under relatively high N concentration (Fig. 1 and 2). As described in the method, we estimated cell size of each phytoplankton group based on forward light scatter of flow cytometry, which was calibrated by 1.75- to 10-$\mu$m standard monodisperse polystyrene beads (Polysciences) (see Fig. S8 in the supplemental material). We are aware of the difficulty in estimating the absolute cell size through forward light scatter (FLS) (32) as well as in volume-to-carbon conversion (33, 34). Acknowledging these limitations, we note that the FLS approach has the advantage of taking into account actual cell size variability compared to the application of a constant carbon per cell factor (35). Despite these constraints, our biomass estimation accounted for approximately half of consumed N (Fig. 1, 2A). This gap may be caused by the limitations of an FLS-based approach. Conversely, uncertainties in size estimation will be more pronounced for the smaller cells than for the larger cells, i.e., *Prochlorococcus* in our experiments, since Mie light scattering theory predicts that scattering per cell volume continues to increase with decreasing particle size. The other possibility of the gap may be due to luxury uptake, (27, 36). Although there are uncertainties in estimating absolute size, which leads to uncertainties in cellular N content, these uncertainties do not change our finding of high N uptake by *Crocosphaera*, known as a bioavailable nitrogen provider. Our samples, filtrate of 1-$\mu$m polypropylene cartridge filter, consisted of up to 5-$\mu$m cells. This shows the leak of phytoplankton through filters (37, 38). Since *Crocosphaera watsonii* (2 to 6 $\mu$m) is larger than other groups (21, 39, 40), higher filtration pressure might lead to underestimation of *Crocosphaera* abundance and biomass compared to other groups. The *Crocosphaera* abundances observed in our experiment are in the same range with nanocyanobacteria, the temporal name for the *Crocosphaera* in reference 25 (see Table 1 in reference 25), which described initial conditions of the same cruise, suggesting that most of the *Crocosphaera* cells went through the filter. During our experiment, biomass in any phytoplankton did not increase at experiment M5 (Fig. 2 and 3). Together with the NH$_4^+$ concentration remaining at more than 30 nM (see Table S1 in the supplemental material), this suggests the non-N-limited nutrient history of the phytoplankton community in experiment M5 (Fig. 3).

The results counter the general image of *Crocosphaera*. It is most known as a diazotroph and is considered to be a provider of N to the environment. Rather, our results support more recent studies, where *Crocosphaera* does not increase the productivity of other phytoplankton (41) or even compete with other species over combined N (23). Surprisingly, our study even shows higher maximum uptake rates of NH$_4^+$ and NO$_3^-$, which allow its dominance just by the uptake of combined N. When the concentration of nitrogen is extremely low, *Crocosphaera* could be outcompeted via N uptake; however, its N$_2$ fixation allows *Crocosphaera* to maintain its biomass at a certain level, which can still be higher than that of nondiazotrophic phytoplankton. This high consumption of NO$_3^-$ may differ from UCYN-A (15, 42–44), which keeps fixing nitrogen under high NO$_3^-$ availability (45, 46), leading to a unique niche acquisition. These results suggest that *Crocosphaera* has high competitiveness under conditions with both low and high nutrients.

Despite this, we generally do not observe the oligotrophic ocean completely dominated by *Crocosphaera*. This may be due to grazing selection. *Crocosphaera* is a unicellular cyanobacterium, a few micrometers to 6 $\mu$m in diameter (47), and its tight coupling with predators has been reported recently (48). The new production of *Crocosphaera* is estimated to support up to 400% of C demand of the main grazers, and the grazing rates of the main predator *Protoperidinium* were found to be nearly equivalent to growth rates of *Crocosphaera* (48).

Conversely, *Trichodesmium*, another N$_2$ fixer in the ocean, is reported to produce a toxin (49–51) and create large colonies of $\sim10^4$ cells (52), potentially protecting themselves from grazing. The other reason might be growth limitation by other nutrients, such as P and Fe. Even though there are reports that *Crocosphaera* shows adaptation for low P and low Fe, having high-affinity phosphate transporters (53) as well as availability of several chemical forms of P (54–56) and high affinity to low Fe concentrations (57) and recycling Fe (14), their relative fitness to such low P or low Fe environments compared to other organisms has not been quantified. Since having nitrogenase enzymes requires a high concentration of Fe, nonnitrogen fixers, such as *Prochlorococcus* and *Synechococcus*, may have lower Fe requirements and are more adapted to Fe depletion. Also, *Crocosphaera* does not seem to fully utilize sulfolipid, which would save P use when compared with that of other cyanobacteria, such as *Synechococcus* (58, 59), and thus may not compete strongly under P limitation. Neither P limitation nor Fe limitation were observed during our observation ($P < 0.05$; repeated measures analysis of variance [RM-ANOVA], *post hoc* Tukey test) (Fig. 2A; see also Fig. S3 and Tables S1, S5, and S6 in the supplemental material).

At the same time, it is largely possible that *Crocosphaera* dominates in some regions in the oligotrophic ocean given its high competitiveness under N limitation, which is the characteristic of the oligotrophic ocean (7, 60). For example, a study of flow cytometry shows a high abundance of *Crocosphaera*-like cells in a wide region of the North Pacific (61), where the abundance of *Trichodesmium* seems limited (62). Also, a recent study shows multiple gene copies in *Trichodesmium* (up to $\sim700$ genes copied per cell) (63), which could overestimate their abundance (64). Given these factors and our analysis showing the high fitness of *Crocosphaera* to both low and high nitrogen concentrations, it is possible that we are still underestimating the relative abundance and role of *Crocosphaera* in global biogeochemical cycling.

## MATERIALS AND METHODS

**Experimental setup and sample collection.** We carried out five macronutrient (N and P)-addition bioassays (M1 to M5) using natural phytoplankton assemblages collected at a station in the subtropical Northwestern Pacific (12°N, 135°E) from 6 to 25 June 2008 during the MR08-02 cruise on the R/V *MIRAI* (see Table S7 in the supplemental material). Water samples were collected from a 10-m depth at 1230 h local time using a Teflon diaphragm pump system. All components of this pump system and associated plastic were washed overnight in a neutral detergent followed by HCl and HNO$_3$, rinsed with heated Milli-Q water, and flushed with seawater for 30 min immediately prior to sample collection. To reduce grazing pressure, we prefiltered seawater from a 10-m depth through an acid-cleaned 1-$\mu$m in-line cartridge filter (Micropore EU; ORGANO) and distributed it into 4-L polycarbonate bottles, which were rinsed overnight in a neutral detergent, followed by 0.3 N HCl, and rinsed with Milli-Q water. We performed three treatments with 100 nM N addition as NaNO$_3$, NH$_4$Cl, or urea. To test P limitation, one treatment with 10 nM NaH$_2$PO$_4$ and our control went without nutrient addition. Forty-five bottles, 5 treatments, and 3 incubation periods (1, 2, or 3 days) were incubated in triplicate for each incubation period (1, 2, or 3 days) on deck in flowthrough seawater tanks covered with a neutral density screen to attenuate light intensity to 50% of its corresponding surface value. Each bottle was used for one time period after washing.

To examine Fe limitation, we carried out three Fe addition bioassays (Fe1 to Fe3) at the same station during the same cruise with the macronutrient addition bioassay experiments (see Table S8 in the supplemental material). Prior to the bioassay experiments, the 2-L polycarbonate incubation bottles had been cleaned according to reference 65. Other polyethylene and Teflon lab wares were cleaned according to reference 66. All washing procedures were carried out in an onshore class 1000 clean air room, and plastic gloves were worn during these operations. To reduce grazing pressure, prefiltrated seawater was prefiltered through a 10-$\mu$m filter of the same manufacturer. The prefiltered water was then dispensed into the corresponding bioassay incubation bottles. Five duplicate treatments were set up as follows: controls without any nutrient addition, phosphate additions with 10 nM NaH$_2$PO$_4$, iron addition with 1 nM FeCl$_3$, an Fe+P treatment with 1 nM FeCl$_3$ and 10 nM NaH$_2$PO$_4$, and Fe+N treatment with an amendment of 1 nM FeCl$_3$ and 100 nM NaNO$_3$. To all treatments containing iron addition, EDTA (1 nM) was added as a buffer. Fe addition treatments were done in an onboard class 100 clean air room. Bottles for the iron addition bioassays were also incubated in on-deck flowthrough seawater tanks covered with neutral density screen to attenuate light intensity to 50% of its corresponding surface value. Iron addition bioassays lasted for 5 days, monitoring total iron (TFe), dissolved iron (DFe), and phytoplankton community composition on days 0, 1, 3, and 5.

**Macronutrient and iron concentrations.** Concentrations of NO$_3^-$ +NO$_2^-$ (N+N), NH$_4^+$, soluble reactive phosphorus (SRP), and urea were measured using a high-sensitivity colorimetric approach with an AutoAnalyzer II (Technicon) and liquid waveguide capillary cells (World Precision Instruments, USA). Triplicate samples for the NO$_3^-$ +NO$_2^-$ (N+N), NH$_4^+$, urea, and soluble reactive phosphorus (SRP) (67, 68) analysis were collected in 100 mL of 0.1 N HCl-rinsed polyethylene bottles. All samples were analyzed onboard, with the exception of urea, which was only measured in the urea treatment. Upon collection, all

samples were stored at −20°C until analysis. We analyzed urea concentrations using the diacetyl monoxime method (69). Detection limits of NO$_3^-$+NO$_2^-$, NH$_4^+$, and SRP were 3, 6, and 3 nM, respectively.

Iron concentrations of the seawater were measured as total iron (TFe), in the whole-water samples collected directly from the pump system, and as dissolved iron (DFe), in the 125 mL of seawater collected in low-density polyethylene bottles (Nalgen; Nalge Nunc International), cleaned according to the methods of reference 65, and filtered through an acid-cleaned 0.22-$\mu$m pore filter (Millipak 100; Millipore). All TFe and DFe samples were acidified with HCl to a pH of <1.5 and stored at room temperature for at least 1 year. Dissolved Fe(III) in seawater samples was determined using catalytic cathodic stripping voltammetry with a detection limit of 6 pM using the approach of reference 70. No contamination during sampling and incubation was detected.

**Flow cytometry.** Flow cytometry (FCM) identified *Prochlorococcus*, *Synechococcus*, picoeukaryotes, and *Crocosphaera* based on cell size and chlorophyll or phycoerythrin fluorescence. Aliquots of 4.5 mL were preserved in glutaraldehyde (1% final concentration), flash-frozen in liquid N$_2$, and stored at −80°C until analysis on land by flow cytometry (PAS-III; Partec, GmbH, Münster, Germany) equipped with a 488-nm argon-ion excitation laser (100 mW). We recorded forward- and side-angle scatter (FSC and SSC), red fluorescence (>630 nm; FL3), and orange fluorescence (570 to 610 nm; FL2). FloMax (Partec, GmbH, Münster, Germany) distinguished *Synechococcus*, *Prochlorococcus*, *Crocosphaera*, and picoeukaryotes based on their autofluorescence properties and their size (61). The instrument settings were standardized for fluorescence intensity and size by using 1.75-, 2.0-, 3.0-, 6.0-, and 10-$\mu$m standard monodisperse polystyrene beads (Polysciences) (see Fig. S8 in the supplemental material).

**Gene analysis.** We collected DNA samples from each treatment of the bioassay and collected aliquots of 0.5 to 1.0 L of sample on 0.2-$\mu$m SUPOR polyethersulfone membrane filters, which we then placed in sterile tubes containing glass beads, froze in liquid N$_2$, and stored at −80°C until further analysis. DNA was extracted according to reference 71 to determine the abundance of *Crocosphaera watsonii* by quantitative PCR (qPCR) using a 5' nuclease assay as described in reference 72.

Quantitative PCR showed that cell densities of FCM-identified *Crocosphaera* were significantly positively correlated with *nifH* gene copies used to quantify the proportion of *Crocosphaera*, indicating that *nifH* abundance accounted for 68% of the variation in FCM-identified *Crocosphaera* ($r^2 = 0.463$, $n = 48$, $P = 0.001$; Pearson product moment correlation). Therefore, this study treated FCM-identified *Crocosphaera* as diazotroph *Crocosphaera*. Cell abundance estimated by qPCR was 0.63 ± 0.23-fold lower than that measured by FCM.

**Nitrogen fixation.** To measure *in situ* N$_2$ fixation activity, we used the acetylene reduction assay of references 73, 74. We dispensed a total of 550-ml bioassay samples into 1,200 mL HCl-rinsed glass polyethylene terephthalate modified with glycol (PETG) bottles with 6 replicates and sealed with butyl rubber stoppers. Aliquots of 120 mL of acetylene (99.9999% [vol/vol]; Koatsu Gas Kogyo, Japan) were injected through the stopper by replacing the same volume of headspace. After 24 h in the on-deck flowthrough seawater tanks, we analyzed ethylene concentrations by converting the ethylene to fixed nitrogen with a molar ratio of 4:1 (75).

**Cellular C and N estimation.** First, we estimated cell size and cell volume based on forward-angle scatter data obtained by flow cytometry following reference 76. Then, we used a conversion factor of 235 fg C $\mu$m$^{-3}$ for *Prochlorococcus*, *Synechococcus*, and *Crocosphaera* (76) to estimate cellular carbon content. For picoeukaryotes, we represented cell volume by converting it into carbon per cell using a modified Strathmann equation (77) as follows:

$$\log C\left(pg/cell\right) = 0.94 \times \log Vol\left(\mu m^3\right) - 0.6$$

Then, using an earlier reported C/N ratio (C/N ratio = 9.1 for *Prochlorococcus*, 8.6 for *Synechococcus*, 8.7 for *Crocosphaera*, 6.6 for picoeukaryotes), we converted the cellular C content into cellular N (21, 78, 79).

**Statistical analysis.** Phytoplankton cell densities of each bioassay were first compared between treatments using repeated measures analysis of variance (RM-ANOVA) with nutrient treatments as a between-subjects factor (5 levels) and time (4 levels) as a within-subjects factor. Treatment effects were considered significant if $P$ was <0.05. Then, the means of five treatments were compared by *post hoc* Tukey test ($n = 3$ replicates per treatment throughout; degrees of freedom = 40). An outlier value in Table S1 in the supplemental material was selected following Smirnov-Grubbs's test ($\alpha = 0.05$).

**Quantitative model of microbial growth.** To quantitatively analyze the fitness of *Crocosphaera* under N-limiting conditions, we ran two simulations. One was to represent the incubation experiment to extract parameters manually, and the other was the simple ecosystem model to simulate their competitiveness under different nutrient concentrations and scenarios. The list of parameters and used values can be found in Tables S4 and S8 in the supplemental material, respectively.

**(i) Simulation of the incubation experiment.** We used the following equations for the growth of phytoplankton to represent the field incubation experiment:

$$\frac{dN_i}{dt} = \mu_i N_i - m_i N_i \tag{1}$$

where $N_i$ (nmol L$^{-1}$) is the cellular nitrogen concentration of phytoplankton $i$ ($i = Cro$, $Oth$: *Crocosphaera* and other phytoplankton, respectively) per volume water, $t$ (d) is time, $\mu_i$ (d$^{-1}$) is the growth rate of phytoplankton $i$, and $m_i$ (d$^{-1}$) is a mortality rate of phytoplankton $i$.

To represent the growth of *Crocosphaera* and other phytoplankton, we used simple growth equations based on the sum of Monod kinetics (80) for each nutrient.

$$\mu_i = V_{Max,i}^{\mathrm{NH4}} \frac{[\mathrm{NH_4^+}]}{[\mathrm{NH_4^+}] + K_i^{\mathrm{NH4}}} + V_{Max,i}^{\mathrm{NO3}} \frac{[\mathrm{NO_3^-}]}{[\mathrm{NO_3^-}] + K_i^{\mathrm{NO3}}} \tag{2}$$

$V_{Max,i}^{\mathrm{NH4}}$ and $V_{Max,i}^{\mathrm{NO3}}$ ($d^{-1}$) are the maximum uptake rate of phytoplankton for $\mathrm{NH_4^+}$ and $\mathrm{NO_3^-}$, respectively, $[j]$ (nmol $L^{-1}$) is the concentration of nutrient $j$ ($j = \mathrm{NO_3^-}$, $\mathrm{NH_4^+}$), and $K_i^{\mathrm{NH4}}$ and $K_i^{\mathrm{NO3}}$ (nmol $L^{-1}$) are the half-saturation constants of nutrient for phytoplankton $i$, respectively. We used the data-fitted quadratic curve of nutrient concentrations (see Fig. S6 in the supplemental material).

**(ii) Simple ecosystem simulation.** To simulate the simple ecosystem situation, we introduced the grazing by zooplankton as follows:

$$\frac{dN_i}{dt} = \mu_i N_i - G_i N_{Zoo} \tag{3}$$

$$\frac{dN_{Zoo}}{dt} = (G_{Cro} + G_{Oth}) N_{Zoo} - m_{Zoo} N_{Zoo}^2 \tag{4}$$

where $G_i$ ($d^{-1}$) is the grazing rate of phytoplankton $i$ by zooplankton, $N_{Zoo}$ (nmol $L^{-1}$) is the nitrogen concentration in zooplankton per volume water, and $m_{Zoo}$ ($d^{-2}$) is the quadratic mortality rate of zooplankton. When we allow nitrogen fixation, we used $\mu_{Cro} = 0.31$ ($d^{-1}$) (a typical growth rate under diazotrophic conditions) (81) if the computation based on equation 2 yielded a value below 0.31 ($d^{-1}$).

For $G_i$ we have applied the KTW method (28) as follows:

$$G_i = G_{max} \left( \frac{N_i^2}{N_{Cro}^2 + N_{Oth}^2} \right) \left( \frac{(N_{Cro} + N_{Oth})^2}{(N_{Cro} + N_{Oth})^2 + K_G^2} \right) \tag{5}$$

where $G_{max}$ ($d^{-1}$) is the maximum grazing rate and $K_G$ (nmol $L^{-1}$) is grazing half saturation. We chose this method since the equation reflects the commonly observed prey-switching behavior of zooplankton (29–31), which stabilizes ecosystems (82, 83). This method allows a diverse phytoplankton to coexist (84) as observed in nature.

**Code availability.** The model developed in this paper has been uploaded in GitHub/Zenodo and is freely available at https://zenodo.org/record/5095790.

## SUPPLEMENTAL MATERIAL

Supplemental material is available online only.
**SUPPLEMENTAL FILE 1**, PDF file, 2.3 MB.

## ACKNOWLEDGMENTS

We thank the captain, crew, and technicians of the R/V *MIRAI* for assistance and support during the research cruise.

This research was financially supported by MEXT grants for Scientific Research on Innovative Areas (24121001 and 24121005 to K. Furuya), JSPS Kakenhi (project 20H03059 to T. Masuda), Czech Research Foundation GAČR (project 20-17627S to O. Prášil and T. Masuda), the Simons Foundation (Life Sciences-Simons Postdoctoral Fellowships in Marine Microbial Ecology, award 544338 to K. Inomura), and the National Science Foundation (NSF) under EPSCoR Cooperative Agreement (OIA-1655221 to K. Inomura).

T. Masuda, K. Furuya, and S. Takeda designed the *in situ* microcosm experiments; T. Masuda, T. Kodama, T. Shiozaki, S. Kitajima, and T. Matsui carried out the experiment and analyzed data supervised by K. Furuya, S. Takeda, M. Sato, and K. Suzuki; T. Masuda and K. Inomura shaped the concept of the study with the supervision of O. Prášil; K. Inomura and G. Armin developed and ran the model. T. Masuda and K. Inomura wrote the original draft with substantial input from all of the authors.

We declare that we have no known competing financial interests or personal relationships that could have appeared to influence the work reported in this paper.

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
