## [Reviewer comments · Microbiology Spectrum]

Microbiology Spectrum

***Crocospaera* as a major consumer of fixed nitrogen**

Takako Masuda, Keisuke Inomura, Taketoshi Kodama, Takuhei Shiozaki, Satoshi Kitajima, Gabrielle Armin, Takato Matsui, Koji Suzuki, Shigenobu Takeda, Mitsuhide Sato, Prášil Ondřej, and Ken Furuya

Corresponding Author(s): Takako Masuda, Japan Fisheries Research and Education Agency

Review Timeline:

Submission Date:	December 2, 2021
Editorial Decision:	February 17, 2022
Revision Received:	June 2, 2022
Accepted:	June 5, 2022

Editor: Jeffrey Gralnick

Reviewer(s): Disclosure of reviewer identity is with reference to reviewer comments included in decision letter(s). The following individuals involved in review of your submission have agreed to reveal their identity: Matthew Peter Clare (Reviewer #1); Alexander B Alleman (Reviewer #2)

Transaction Report:

DOI: <https://doi.org/10.1128/spectrum.02177-21>

February 17, 2022

Dr. Takako Masuda
The Czech Academy of Sciences
Třeboň
Czech Republic

Re: Spectrum02177-21 (*Crocospaera* as a major consumer of fixed nitrogen despite its capability of nitrogen fixation)

Dear Dr. Takako Masuda:

Thank you for submitting your manuscript to Microbiology Spectrum. There remain two main issues with the manuscript, which I will summarize. The first is a grammatical one, raised by both reviewers. Reviewer 1 has provided a detailed list of specific comments related to this. The second issue relates to making sure the claims are fully supported by the data presented - a founding principle of Spectrum. In this regard I don't expect any new experimentation to be provided, rather, some of the claims may need to be reviewed, revised and possibly 'toned down'. Please pay careful attention to this in your revision.

Also, Reviewer 2 raised an issue regarding the origin of the samples (referring to some potential overlap with a preprint from your group here: <https://www.researchsquare.com/article/rs-394978/v1>) - this doesn't seem like a major issue to me, but would benefit from some clarification.

Link Not Available

Sincerely,

Jeffrey Gralnick

Journals Department
Reviewer comments:

Reviewer #1 (Comments for the Author):

I have made a series of comments in an attached document.

I think that the major issue is that the paper has a series of grammatical and spelling errors that need review.

I have listed three pages of them in the document that I have attached as well as my thoughts on the manuscript.

Reviewer #2 - Please see the attached document

Staff Comments:

Preparing Revision Guidelines

Please return the manuscript within 60 days; if you cannot complete the modification within this time period, please contact me. If you do not wish to modify the manuscript and prefer to submit it to another journal, please notify me of your decision immediately so that the manuscript may be formally withdrawn from consideration by Microbiology Spectrum.

This manuscript presents the results obtained from a series of open-ocean surface water communities regarding nitrogen and ammonium consumption. I appreciate the time taken to work on this and respond to the original two reviewers. The efforts there are to be commended.

Grammatical errors:

Please see the following points for editing in order to improve the readability of the proposed paper:

Line 45: “N limiting” should be “N-limiting”.

Line 45: “...oceans, and thus heightens its...” is clunky and could be better written as “...oceans heightening its...”

Line 49: “has been known as” is a particularly clunky phrase to use in this sentence. Consider “is”

Line 49: There is no need for the word “the” to be present and this slows down the reading of the sentence. Consider “...has been known as a key nitrogen (N) supplier in marine ecosystems...”

Line 50: “since it has been estimated to” is clunky; I feel that you are trying to say, “and it has been estimated to contribute up to...”

Line 50: “...contribute up to about a half” is not a polished phrase and should be rewritten. Please reconsider how to phrase this. Perhaps, “...contribute up to half of oceanic...”. This should also be referenced.

Line 53: “A recent study” should be referenced.

Line 53: “Yet, how *C.* competes for combined-N is missing” is a clunky sentence and could be better phrased as “Despite its importance in nitrogen cycling, the methods by which *C.* compete are not fully understood currently”.

Line 70: “N₂ fixing” should be “N₂-fixing”.

Line 88-90: This sentence, starting “Initially”, should be rewritten or broken down in order to improve readability.

Line 108: “N₂ fixation” should be “N₂-fixation”.

Line 121: I am not convinced of the need for the “1)” and “2)” in the format presented. These can be deleted, or the sentence should be converted into a list format. My personal preference would be for deletion.

Line 126: “Modeling” should be spelled as “Modelling”.

Line 129: Is “Especially” the correct word to use here?

Line 129: “About” does not feel appropriate. Please provide specific details.

Line 136: “Here, we simulate...” should be “Here, we simulated...”. The tenses have been confused. *i.e.* in line 128, “we simulated”

Line 136: “use” should be “used”. The tenses have been confused. *i.e.*, in line 128, “we simulated”. Similar occurrences of tense-confusion may have arisen throughout this paper; however, I may not have been able to list all tense-confusion examples here. I would encourage the authors to check this before publication.

Line 137: “because” is better in this specific case than “since” for readability.

Line 153: A comma should be inserted after the “(30)”, as below:

“...through forward light scatter (FLS) (30), as well as in volume to carbon conversion”.

Line 153: “volume to carbon” should be hyphenated, as below:

“Volume-to-carbon”.

Line 153: “In fact,” is a very informal opening to a sentence and is not required here.

Perhaps this could be rewritten in this manner:

“Despite these constraints, our biomass estimated accounted for approximately half of consumed N”. I have changed “about” to “Approximately” as well.

Line 154-155: This sentence is not grammatically appropriate and should be rewritten as below:

“This gap may be caused by the limitations of an FLS-based approach”

Line 155: “On the other hand,” is a very informal opening to a sentence and could be more professionally written as “However” or “Conversely”.

Line 156: “therefor” should be spelt with an “e” on the end. *i.e.*, “therefore”.

Line 158: “possibly” should probably be replaced by the word “possibility”. I would ask the authors to check this.

Line 159: “limitation” should be replaced by “limitations”.

Line 162: “don’t” should be rewritten as “do not” for a more formal reading of this work and to be consistent with the remainder of the paper where the contraction has not been used.

Line 167: I feel this sentence is best being split into two sentences. Please consider the following:

“The results counter the general image of *Crocospaera*. It is most known as a diazotroph and is considered to be a provider of N to the environment. Rather, our results...”

Line 172: “just by” should be written as “just by the”.

Line 172: “When nitrogen concentration is extremely low,...” should be written as “When the concentration of nitrogen is extremely low”.

Line 173: I would avoid using “they” when you are writing a journal as you are doing. Use the name of the thing that you are talking about. *i.e.*, “*Crocospaera*”.

Line 173: Consider rewriting this sentence as below:

“...could be outcompeted via N uptake; however, their N₂-fixation...”

Line 180: “One reason might be the grazing selection” should be rewritten to “This may be due to grazing selection”.

Line 181: The description of the size of the *Crocospaera* is a subclause and should be enveloped in commas. *i.e.*, “...is a unicellular cyanobacterium, a few....in diameter (47), and its tight coupling with predators...”.

Line 182: “is” should be written as “has been”. There has been some tense confusion here.

Line 185: “On the other hand,” should be “Conversely,”.

Line 185: “N₂ fixer” should be “N₂-fixer”.

Line 185-187: This sentence has a lot of commas and sub-clauses and should probably be broken up. This should be rewritten to improve readability.

Line 187: “Another reason...” feels like the start of a new idea and a new paragraph.

Line 193: “Fe depletion” should be “Fe-depletion”

Line 194: “as opposed to” should be “when compared with”

Line 194: “P use” should be “P-use”

Line 195: “P limitation” should be “P-limitation” on both occasions (and throughout the document).

Line 198: “N limitation” should be “N-limitation”.

Line 202: “would” should be “could” unless you are saying that this overestimation is an absolute fact. Some hedging here is probably best.

Line 203: “Given these factors and...” should be better written as “Given these factors, and our analysis showing...”.

Line 203: “their” should not be used here. Write the name of the thing you are referring to.

Line 258: “N₂ fixation” should be “N₂-fixation”

Line 280: “Then, means between five treatments were...” could be written as “Then, the means of five treatments were...”

Line 284: “N limiting” should be “N-limiting”.

Line 305: “nitrogen fixation” should be “nitrogen-fixation”.

Content:

I feel this paper is of value to the scientific community and the efforts to imitate ocean ecosystems through these experiments are commendable. I do feel that more is needed in the methodology section to properly explain this. I feel that this is the novelty of the work and needs further highlighting, building on that which has been done in the response to the second reviewer. This should be detailed meticulously in the methodology and is not from that which I can see.

I feel the response to the first reviewer’s first question is excellent, as is the question. This could be developed more in the paper, particularly the calibrations used, which would be questioned by anyone reading this paper. This should be included as the authors have done in the response to the reviewer.

The response to the second question from the first reviewer is good and this kind of justification, as is included in the paper, could be fleshed out somewhat; however, this does appear to be a good justification.

I agree with the first reviewer in their third point that the justification for ignoring results M1,2,4,5 was weak. I think that this has been improved with the explanation provided and the insert into the manuscript.

I agree with the first reviewer in their fourth point that the discussion around P (and Fe later) has not been fully explored. I do feel that more work should have been done initially with the samples in order to analyse for any P- or Fe-containing molecules (or other molecules we understand are important for the limiting or boom of a population). I feel this is a fundamental weakness holding this paper back and would have been a true highlight if this manuscript was to analyse for more than simply Nitrogen.

I feel that the second reviewer is perhaps airing on the side of strictness when they say that they find the paper lacking in novelty. I feel that the comment about completeness may well have been correct based on the responses provided. I feel the paper is much more complete thanks to the additional information provided in response to the previous reviewers' comments. I feel the discussion in lines 212-219 are vital to the understanding of this work. Similarly, the addition of lines 162-166 and 85-90 are important as well for the procedures followed. I feel these additions very much help the completeness of this work.

I agree with the reviewer's comments regarding C. taking up ammonium and nitrate and that this would occur at higher rates when it could. This is not novel scientific theory being presented in the manuscript; however, I feel that the attempts to replicate the real-world are where the value of this manuscript lie. More information could be added to explain the conditions that the samples were kept at during analysis and this would be appreciated by those that view the real-world replication as the importance. I feel this would very much help both the understanding of the work and the replicability of the work.

I agree with the second reviewer's comment with regard to the use of the KiW model, and the explanation is reasonable.

Overall:

I have provided a list of grammatical and spelling errors that I feel must be made in order to publish this paper.

I feel that the most interesting part of this work, being the almost *in situ* nature of this work and the attempted replication of real-world conditions, is not fully described in the methodology and could be heavily fleshed out. I am interested currently about the incubation conditions on deck and would be interested to know more about those.

I also feel that further analysis of the initial samples for more than Nitrogen-based molecules would have provided this work with a much better standing. This is especially true as the paper discusses the potential roles of P and Fe, almost as an afterthought, in microbial marine communities. I would have liked to see this being discussed further and would have made this paper of more use to more.

Despite these comments, I feel this paper is of value. I do not feel that it is publishable in its current form and the amendments listed above to the grammar and spelling must be made before this work is considered again. I have tried to catch all errors, but I would advise another proof-read of the work in order to ensure that it is as well-written as possible.

Reviewer #1

This manuscript presents the results obtained from a series of open-ocean surface water communities regarding nitrogen and ammonium consumption. I appreciate the time taken to work on this and respond to the original two reviewers. The efforts there are to be commended.

We appreciate the time and effort devoted to reviewing and polishing up our work. We followed most of the comments, which improved the manuscript significantly. We further revised the manuscript and provided 7 extra supplemental Tables.

1. We revised grammatical errors following suggestions.
2. To clarify the initial water mass changed between experiments M3 and 4., we provided chemical and physical conditions as Table S1.
3. To clarify the experimental procedures, we revised experimental settings and sample collection and included the procedure of Fe enrichment experiments. We provide Table S 7 and 8 as summaries of the procedure.

Thanks to the reviewer's comments, we believe that the manuscript has improved significantly, and hope that the revision will be found sufficient. We describe our response to each point below:

Grammatical errors:

Please see the following points for editing in order to improve the readability of the proposed paper:

Line 45: "N limiting" should be "N-limiting".

Line 45: "...oceans, and thus heightens its..." is clunky and could be better written as "...oceans heightening its..."

Line 49: "has been known as" is a particularly clunky phrase to use in this sentence. Consider "is"

Line 49: There is no need for the word "the" to be present and this slows down the reading of the sentence. Consider "...has been known as a key nitrogen (N) supplier in marine ecosystems..."

Line 50: "since it has been estimated to" is clunky; I feel that you are trying to say, "and it has been estimated to contribute up to..."

Line 50: "...contribute up to about a half" is not a polished phrase and should be rewritten. Please reconsider how to phrase this. Perhaps, "...contribute up to half of oceanic...". This should also be referenced.

Line 53: "A recent study" should be referenced.

Line 53: "Yet, how *C.* competes for combined-N is missing" is a clunky sentence and could be better phrased as "Despite its importance in nitrogen cycling, the methods by which *C.* compete are not fully understood currently".

Line 70: "N₂ fixing" should be "N₂-fixing".

Line 88-90: This sentence, starting "Initially", should be rewritten or broken down in order to improve readability.

Line 108: "N₂ fixation" should be "N₂-fixation".

Line 121: I am not convinced of the need for the "1)" and "2)" in the format presented. These can be deleted, or the sentence should be converted into a list format. My personal preference would be for deletion.

Line 126: "Modeling" should be spelled as "Modelling".

Line 129: Is "Especially" the correct word to use here?

Line 129: "About" does not feel appropriate. Please provide specific details.

Yes, here we put the specific value and removed "about."

Line 136: "Here, we simulate..." should be "Here, we simulated...". The tenses have been confused. *i.e.* in line 128, "we simulated"

Line 136: “use” should be “used”. The tenses have been confused. *i.e.*, in line 128, “we simulated”. Similar occurrences of tense-confusion may have arisen throughout this paper; however, I may not have been able to list all tense-confusion examples here. I would encourage the authors to check this before publication.

Line 137: “because” is better in this specific case than “since” for readability.

Line 153: A comma should be inserted after the “(30)”, as below:

“...through forward light scatter (FLS) (30), as well as in volume to carbon conversion”.

Line 153: “volume to carbon” should be hyphenated, as below:

“Volume-to-carbon”.

Line 153: “In fact,” is a very informal opening to a sentence and is not required here. Perhaps this could be rewritten in this manner:

“Despite these constraints, our biomass estimated accounted for approximately half of consumed N”. I have changed “about” to “Approximately” as well.

Line 154-155: This sentence is not grammatically appropriate and should be rewritten as below:

“This gap may be caused by the limitations of an FLS-based approach”

Line 155: “On the other hand,” is a very informal opening to a sentence and could be more professionally written as “However” or “Conversely”.

Line 156: “therefor” should be spelt with an “e” on the end. *i.e.*, “therefore”.

Line 158: “possibly” should probably be replaced by the word “possibility”. I would ask the authors to check this.

Line 159: “limitation” should be replaced by “limitations”.

Line 162: “don’t” should be rewritten as “do not” for a more formal reading of this work and to be consistent with the remainder of the paper where the contraction has not been used.

Line 172: “just by” should be written as “just by the”.

Line 172: “When nitrogen concentration is extremely low,...” should be written as “When the concentration of nitrogen is extremely low”.

Line 173: I would avoid using “they” when you are writing a journal as you are doing. Use the name of the thing that you are talking about. *i.e.*, “*Crocospaera*”.

Line 173: Consider rewriting this sentence as below:

“...could be outcompeted via N uptake; however, their N₂-fixation...”

Line 180: “One reason might be the grazing selection” should be rewritten to “This may be due to grazing selection”.

Line 181: The description of the size of the *Crocospaera* is a subclause and should be enveloped in commas. *i.e.*, “...is a unicellular cyanobacterium, a few....in diameter (47), and its tight coupling with predators....”.

Line 182: “is” should be written as “has been”. There has been some tense confusion here.

Line 185: “On the other hand,” should be “Conversely,”.

Line 185: “N₂ fixer” should be “N₂-fixer”.

Line 185-187: This sentence has a lot of commas and sub-clauses and should probably be broken up. This should be rewritten to improve readability.

We have shortened and rearranged the sentence for clarity.

Line 187: “Another reason...” feels like the start of a new idea and a new paragraph.

Line 193: “Fe depletion” should be “Fe-depletion”

Line 194: “as opposed to” should be “when compared with”

Line 194: “P use” should be “P-use”

Line 195: "P limitation" should be "P-limitation" on both occasions (and throughout the document).

Line 198: "N limitation" should be "N-limitation".

Line 202: "would" should be "could" unless you are saying that this overestimation is an absolute fact. Some hedging here is probably best.

Line 203: "Given these factors and..." should be better written as "Given these factors, and our analysis showing...".

Line 203: "their" should not be used here. Write the name of the thing you are referring to.

Line 258: "N2 fixation" should be "N2-fixation"

Line 280: "Then, means between five treatments were..." could be written as "Then, the means of five treatments were..."

Line 284: "N limiting" should be "N-limiting".

Line 305: "nitrogen fixation" should be "nitrogen-fixation".

Thank you very much for correcting the grammar issues. We revised the text following the suggestions, except adding references in the "Importance" section, since reference is not expected in the section.

Content:

I feel this paper is of value to the scientific community and the efforts to imitate ocean ecosystems through these experiments are commendable. I do feel that more is needed in the methodology section to properly explain this. I feel that this is the novelty of the work and needs further highlighting, building on that which has been done in the response to the second reviewer. This should be detailed meticulously in the methodology and is not from that which I can see.

Thank you very much for recognizing the value of the work. Following the suggestion, we have revised methodology part to clarify the experimental procedure. In addition, now we explained how we measured iron concentrations (Lines 235-241, 242-258, 270-278).

I feel the response to the first reviewer's first question is excellent, as is the question. This could be developed more in the paper, particularly the calibrations used, which would be questioned by anyone reading this paper. This should be included as the authors have done in the response to the reviewer.

To clarify the potential problem of size estimation by flow cytometry, we added the sentence explaining the phytoplankton size calibration method, and relocated the sentence related to limitations of other method; constant carbon per cell factor (Lines 156-162):

"Our study shows high uptake of N by *Crocospaera* under relatively high N concentration (Figs 1, 2). As described in the method, we estimated cell size of each phytoplankton group based on forward light scatter of flow cytometry, which was calibrated by 1.75 to 10 μ m standard monodisperse polystyrene beads (Polysciences) (Fig. S8). We are aware of the difficulty in estimating the absolute cell size through forward light scatter (FLS) (Ackleson and Spinrad, 1988), as well as in volume-to-carbon conversion (Campbell et al., 1994, Zubkov et al., 1998). Acknowledging the limitations, we note that the FLS approach has the advantage of taking into account actual cell size variability, compared to the application of a constant carbon per cell factor (Shalapyonok et al., 2001)."

-----Related reference-----

Ackleson SG, Spinrad RW. 1988. Size refractive index of individual marine particulates: a flow cytometric approach. *Applied Optics* 27:1270 - 1277.

Campbell L, Nolla HA, Vaultot D. 1994. The importance of *Prochlorococcus* to community structure in the central North Pacific Ocean. *Limnol Oceanogr*, 39:954-961.

Zubkov MV, Sleight MA, Tarran GA, Burkill PH, Leakey RJG. 1998. Picoplanktonic community structure on an Atlantic transect from 50 degrees N to 50 degrees S. *Deep-Sea Research Part I-Oceanographic Research Papers* 45:1339-1355.

Shalapyonok A, Olson RJ, Shalapyonok LS. 2001. Arabian Sea phytoplankton during Southwest and Northeast monsoons 1995: composition, size structure and biomass from individual cell properties measured by flow cytometry. *Deep-Sea Research Part II-Topical Studies in Oceanography* 48:1231-1261.

The response to the second question from the first reviewer is good and this kind of justification, as is included in the paper, could be fleshed out somewhat; however, this does appear to be a good justification.

To clarify our point, we further revised the text as following (Lines 169-171):

"Although there are uncertainties in estimating absolute size, which leads uncertainties in cellular N content, these uncertainties do not change our finding of high N uptake by *Crocospaera*, known as a bioavailable nitrogen provider."

I agree with the first reviewer in their third point that the justification for ignoring results M1,2,4,5 was weak. I think that this has been improved with the explanation provided and the insert into the manuscript.

Thank you for the positive feedback.

I agree with the first reviewer in their fourth point that the discussion around P (and Fe later) has not been fully explored. I do feel that more work should have been done initially with the samples in order to analyse for any P- or Fe-containing molecules (or other molecules we understand are important for the limiting or boom of a population). I feel this is a fundamental weakness holding this paper back and would have been a true highlight if this manuscript was to analyse for more than simply Nitrogen.

Now we provided the initial physical, chemical and biological conditions including phosphorus and iron concentrations in Table S1.

In addition, to clarify how we have examined P- and Fe- limitation, we have revised description of "Experimental setup and sample collection" in Materials and methods (Lines 235-241, 242-258), and provided summary of the experimental procedures in Tables S7 and S8.

We added detailed information on adaptation for low P and low Fe known for *Crocospaera* in Lines 201-206.

I feel that the second reviewer is perhaps airing on the side of strictness when they say that they find the paper lacking in novelty. I feel that the comment about completeness may well have been correct based on the responses provided. I feel the paper is much more complete thanks to the additional information provided in response to the previous reviewers' comments. I feel the discussion in lines 212-219 are vital to the understanding of this work. Similarly, the addition of lines 162-166 and 85-90 are important as well for the procedures followed. I feel these additions very much help the completeness of this work.

I agree with the reviewer's comments regarding C. taking up ammonium and nitrate and that this would occur at higher rates when it could. This is not novel scientific theory being presented in the manuscript; however, I feel that the attempts to replicate the real-world are where the value of this manuscript lie. More information could be added to explain the conditions that the samples were kept at during analysis and this would be appreciated by those that view the real-world replication as the

importance. I feel this would very much help both the understanding of the work and the replicability of the work.

Thank you for your positive feedback. As above, we now included initial conditions of the experiments in Table S1.

Now we provide detail method on the nutrient and Fe analysis including how to keep samples until analysis (Lines 263-267, 270-278).

I agree with the second reviewer's comment with regard to the use of the KiW model, and the explanation is reasonable.

Overall:

I have provided a list of grammatical and spelling errors that I feel must be made in order to publish this paper.

I feel that the most interesting part of this work, being the almost *in situ* nature of this work and the attempted replication of real-world conditions, is not fully described in the methodology and could be heavily fleshed out. I am interested currently about the incubation conditions on deck and would be interested to know more about those.

I also feel that further analysis of the initial samples for more than Nitrogen-based molecules would have provided this work with a much better standing. This is especially true as the paper discusses the potential roles of P and Fe, almost as an afterthought, in microbial marine communities. I would have liked to see this being discussed further and would have made this paper of more use to more.

Despite these comments, I feel this paper is of value. I do not feel that it is publishable in its current form and the amendments listed above to the grammar and spelling must be made before this work is considered again. I have tried to catch all errors, but I would advise another proof-read of the work in order to ensure that it is as well-written as possible.

Thank you for your detailed review, and constructive comments. We hope that our revision will be found satisfactory.

Reviewer 2:

The manuscript, "Crocospaera as a major consumer of fixed nitrogen despite its capability of nitrogen fixation", details *in situ* microcosm incubations under oligotrophic growth conditions. The authors present evidence and a simple model showing how the diazotrophic Crocospaera has high DIN uptake rates making it competitive at high and low nitrogen conditions. While having robust quantitative data in these oligotrophic environments is extremely important for more extensive ocean modeling, I feel the authors' claims are not supported by the data they provide. Significant revisions to the interpretation of the data in the other experiments must be presented to support the claims made.

We appreciate the time and effort devoted to reviewing and polishing up our work. Following the reviewer's comments, we further revised the manuscript. The key revisions include:

1. We changed the title of the manuscript
2. To clarify the experimental procedure, we revised the text in experimental setup and sample collection and included an Fe enrichment experiment. We provided summaries in Tables S 7 and 8.
3. To clarify the effect of N enrichment, we provided the statistical results in Tables S5 and S6.

We describe our response to each point below:

Major revisions:

1) The title and some of the introduction are worded to suggest diazotrophs do not consume DIN or at least do not actively compete for DIN. In contrast, the literature cited and the manuscript results suggest that this is not the case. "Crocospaera as a major consumer of fixed nitrogen **despite** its capability of nitrogen fixation" seems like a binary while literature supports simultaneous nitrogen fixation and ammonia consumption. The title needs to be rearranged or reworded, and the introduction can go into more details of Dekaezemacker and Bonnet (2011) and the authors Masuda et al. (2013). This will allow the introduction to set up the experiments presented, repeating these results in an *in situ* experiment.

To avoid the confusion, we have changed the title from "*Crocospaera* as a major consumer of fixed nitrogen despite its capability of nitrogen fixation" to "*Crocospaera* as a major consumer of fixed nitrogen". (Line 1)

Following the suggestion, we now present more detail understanding of Dekaezemacker and Bonnet (2011) and Masuda et al., 2013 (Lines 72-79):

"Earlier studies examined the effect of combined N, such as ammonium (NH_4^+) and nitrate (NO_3^-) on metabolic activities, and reveal *Crocospaera*'s ability to assimilate combined N (Dekaezemacker and Bonnet 2011, Masuda et al., 2013). As reported from *Trichodesmium* (Holl and Montoya, 2005), increasing concentrations of NH_4^+ enrichment increases NH_4^+ uptake activities and inhibits N_2 fixation rates up to ~80% (Dekaezemacker and Bonnet 2011, Masuda et al., 2013), while NO_3^- enrichment did not inhibit N_2 fixation rate at any of the tested NO_3^- concentrations (up to 10 μM) (Dekaezemacker and Bonnet 2011). When remaining combined N concentrations in the cultures are at a nanomolar level, *Crocospaera* kept fixing N_2 (Dekaezemacker and Bonnet 2011, Masuda et al., 2013). "

-----Related reference-----

Dekaezemacker J, Bonnet S. 2011. Sensitivity of N_2 fixation to combined nitrogen forms (NO_3^- and NH_4^+) in two strains of the marine diazotroph *Crocospaera watsonii* (Cyanobacteria). *Mar Ecol Prog Ser* 438:33-46.

Masuda T, Furuya K, Kodama T, Takeda S, Harrison PJ. 2013. Ammonium uptake and dinitrogen fixation by the unicellular nanocyanobacterium *Crocospaera watsoni* in nitrogen-limited continuous cultures. *Limnol Oceanogr* 58:2029–2036.

Holl CM, Montoya JP. 2005. Interactions between nitrate uptake and nitrogen fixation in continuous cultures of the marine diazotroph *Trichodesmium* (Cyanobacteria)¹. *Journal of Phycology* 41:1178-1183.

3) The description of the experimental setup was very confusing and made the results hard to understand for the reader. A description or a diagram of the whole experiment protocol from start to finish (for a single M experiment) might help.

To clarify the scheme of the experiment, we revised "Experimental setup and sample collection".

Now we provided detailed description of "Experimental setup and sample collection" in Materials and methods (Lines 225-241, 242-258), and provided summary of the experimental procedures in Tables S7 and S8.

Line 219 – Was the 4-L bottle a single sample in triplicate (so 3x 4L bottles for every condition)?

Yes, 3 bottles for each condition. This is informed in the "Experimental setup and sample collection" as above (Lines 237-238).

"Forty-five bottles and 5 treatments and 3 incubation period; 1, 2, or 3 days in triplicate were incubated for each incubation periods (1, 2, or 3 days)"

Line 218 – This filtration step is still unclear even though the authors have addressed it on Line 164 in the discussion. The primary organism of study in this manuscript, *Crocospaera* is much larger than the 1µM filter used to obtain the sample. The explanation given to a previous reviewer for seeing *Crocospaera* in the sample was to claim a leaky filter (see the response to reviewer #2). However, all this means is that the authors filter out 99% of the *Crocospaera* in the original seawater sample. At least this should be clearer as making any hypothesis on the role of *Crocospaera* should be taken lightly under these experimental circumstances.

We can only assume that the quality test for the filter was done with the standard beads, which does not change its shape. However, in case of cyanobacteria, it has flexibility with its shape. We don't know the accurate capture efficiency, but the *Crocospaera* abundances observed in our experiment were in same range with those of the Nanocyanobacteria - the temporal name for the *Crosoaphaera* in Shiozaki et al., 2013 (Table 1), which described initial conditions of the same cruise - suggesting that most of *Crocospaera* leaked the filter. We discuss this in the discussion (Lines 174-177).

Line 90-92: Where are these measurements coming from? Which day were they taken? Why do they not match any data in Table S2?

These values are the initial nutrient concentration of seawater without any treatment. Previous Table S2, current Table 3 showed the concentration collected for each bioassay experiment, which shows the effect of filtration (Table S3). Now we show present initial nutrient concentrations, without any treatment in Table S1 (Line 174-180).

Line 93: Is this the same experiment as described in the preprint (24), as in the same bottles sampled, or were these two parallel experiments in separate bottles? Honesty about these two experiments and their differences must be presented in both manuscripts.

Yes, these are the same experiment. Since we presented initial conditions in this manuscript, we will refer this manuscript in the preprint, which will be published later. Therefore, we now describe the results in Table S1 (Lines 91-96).

2) One of the major concerns in the manuscript is the significant variance in results through experiments M1-M5 and the lack of clarity and specification for choosing the M3 experiment. This leads to the final analysis, model building, and hypothesis generation on one single experiment while ignoring contradictory results in the others. The authors chose M3 by criteria that the M3 is the only experiment with low initial nutrient conditions (<20nM) and low biomass (<250 cells). Is there any other literature support for this criteria? Even with other literature to support this distinction and selection of M3, the variance in measurement would still need to be considered. For example, M2 was not selected due to high initial biomass (Line 124), but the biomass is 270 +/- 225! Error that large needs to be considered in the criteria and does not justify only looking at M3. The same situation can be applied to experiment M1 with 22 nM of NO₂/NO₃ just narrowly out of the criteria range, but the experiment results contradict the M3 experiments.

To the best of our knowledge, only Glover et al., (2007) reported an effect of nutrient enrichment at the nanomolar levels on physiology of the *in situ* phytoplankton assemblage. Glover et al., (2007) reported the "strong inverse relationship between the degree of nitrate-enhanced photosynthesis and the ambient concentration from which the organisms were taken", as well as "picoplankton growing at < 60 nM nitrate rapidly responded to nanomolar nitrate enrichment with luxury consumption and enhanced photosynthesis in proportion to their ambient nitrate environment." Our observation confirms the observation of Glover et al., (2007). They further describe the importance of nutrient history of phytoplankton (Droop et al., 1974). Confirming the observation of Glover (2007), we expect the clearest response to nutrient enrichment from the most nutrient-starved phytoplankton community, which is observed at Experiment M3.

To clarify the reason why we selected M3, we have revised the sentences as following (Lines 123-129).

" The ideal conditions to test relaxation from nutrient stress are to use a nutrient starved phytoplankton community, which spends a long time under low nutrient conditions (Droop et al., 1974, Glover et al., 2007). Considering the nutrient history of the *in situ* phytoplankton community, we selected M3 (Tables S1 and S3) to be the best example to observe relief from nutrient stress, since the water mass changed from more N-limited water to N-rich between M3 and M4. The N-limited nutrient history of phytoplankton was confirmed by low initial nutrient conditions and low biomass of the targeted organisms. Thus, we focus on the data from experiment M3 for modelling analysis. "

We have recalculated the biomass for experiment M2 by removing the one data point, which was more than 4 times higher than the other two samples, following Smirnov-Grubbs's test for outliers ($n=3$, $t = 1.41$, $\alpha = 0.05$). As a consequence, initial *Crocospaera* biomass in M2 changed from 270 ± 225 to 129 (Table S1). Standard deviation of 22 nM for NO₃⁻ + NO₂⁻ concentrations is acceptably low. The reason why we want to focus on M3 is not because of these values, but because of nutrient history of phytoplankton community.

-----Related references-----

Droop MR. 1974. The nutrient status of algal cells in continuous culture. J Mar Biolog Assoc U K 54:825-855.

Glover HE, Garside C, Trees CC. 2007. Physiological responses of Sargasso Sea picoplankton to nanomolar nitrate perturbations. J Plankton Res 29:263-274.

4) Using conversion factors and not cell count might lead to misleading or confusing data:

Figure 2 B and C , Figure S4– These plots are unclear and need more explanation or be redone. The nitrogen and the carbon were estimated with a conversion factor plotting them in a xy plot does nothing other than show the conversion factor.

How was the NH₄ vs. NO₂/3 contribution calculated? Was this biomass increase in the NH₄ or NO₃/2 condition compared to control conditions? If so, can you explain why M5 or M4 >75% of the contribution to N is from NH₄, but there is little difference in growth compared to the control condition with no added N?

These plots bring little to the discussion and would rather have a cell count or other flow cytometry data.

We believe that the biomass is the best way to estimate contribution of C- or N-content for each phytoplankton group, since cell size plays an important role in estimation of cellular C- and N- content. Even though there are limitations in FLS-based approach, the method does not overestimate cellular C- or N- content of *Crocospaera*. Since *Crocospaera* is about > 3 times larger than other observed phytoplankton groups, *Crocospaera* require more molecules including C and N. This is not because of conversion factor, but because of cell size.

The contributions of NH₄⁺ and NO₃⁺ are calculated from NH₄⁺ or from NO₃⁺ enrichments, respectively. This is now explained in the caption of Fig. 2 and Fig. S4.

"The contributions of NH₄⁺ - or NO₃⁺ - N were estimated from either NH₄⁺ or NO₃⁺ enrichment. " (Lines 394-395)

The biomass of *Crocospaera* didn't always increase in NH₄⁺ or NO₃⁺ enrichment including experiments M4 and M5 (Table S6). Biomass of *Prochlorococcus* and pico-eukaryotes increased by NH₄⁺ and urea addition in all the experiments while *Synechococcus* increased by NO₃⁺ addition, except for Ex M5. The growth stimulation of nitrogen enrichment was weaker when initial N concentration was higher, which likely reflects nutrient history of the phytoplankton community.

By the bubble plot in Fig. 2B, C and Fig. S4, we would like to show the significant contribution of *Crocospaera* to C- and N- biomass compared to the other phytoplankton groups. Thus, we consider it reasonably valid to keep them in the manuscript.

Other flow cytometry data may help describing cellular pigments such as chlorophyll (red fluorescence) or phycoerythrin (orange fluorescence), but unfortunately, we have not examined the way to convert fluorescence intensity to a pigment content.

Minor revisions

Line 74 – I would not use the word recent to describe these studies from 2011 and 2013.

Changed from "Recent" to "Earlier" (Line 72).

Line 105 – How was the switch from oligotrophic to mixed-water determined. What values in carbon, nitrogen, cell counts changed?

Excuse us for missing information. Now we present physical, chemical and biological data of initial water in Table S1. And the results were described in the results (Lines 91-96).

Line 122 – Table S2 AND Table S1 (for cell counts)

Revised accordingly.

Line 133 – Please cite the claim that "predicted maximum NO₃ uptake rate for *Crocospaera* is also higher than for other phytoplankton..."

We now refer to Table S4 and added a specific value (Lines 132-135).

Line 146 – Figure S7C, Is there no difference between other phytoplankton and Croco in nitrogen-fixing conditions? Or is this a mistake?

We understand the reviewer's comment. We believe that the figure and the statement in the original manuscript are correct because once we include N₂ fixation, *Crocospaera*'s population is higher than other phytoplankton. This may not be easily seen in Figure S7C because it is plotted in log scale. Thus, we added a panel in a linear scale for clarification (Fig. S7D). Accordingly, we have referred to Fig. S7D in the main text (Line 174).

Fig. S7 Simulated transition of cellular N in a simple ecosystem model for three different scenarios. (A) The concentrations for NH₄⁺ and NO₃⁻ are both 100 nmol L⁻¹. (B)(C) The concentrations for NH₄⁺ and NO₃⁻ are both 1 nmol L⁻¹. In only (C) *Crocospaera* may acquire N via N₂ fixation. (D) The same results as (C) but plotted in a linear scale and different axis ranges. Croco: *Crocospaera*. Other: other phytoplankton. Parameters are based on NO₃⁻ added case.

Line 159 – Define luxury uptake in the text.

We cited Droop et al., 1974. (Line 169)

Line 202 – change copies "of" to copies "in"

Revised.

Line 210 – No journal or arxiv doi link in ref 24

Instead of referring to an unpublished paper, now we described the detail procedure of the experimental set up and sample collections including Table S7 (Line 228).

Line 245 – Fe addition bioassay?

Now we explained the procedure of Fe addition bioassay (Lines 241-257).

Line 338 – Give full names for Pro, Syn, etc.

Line 338 and lines around 338 in original don't contain "Pro" nor "Syn", or spelled out. Caption for Fig. 2 contains full names for Pro, Syn, Cro, PicoE (Lines 396-397).

Figure 1 – Lines are too thin, and colors are too similar to differentiate the lower values. Maybe a log scale can help? Figure S1 is even worse

We changed the thickness and colour of the lines. To keep message simple as 'there was no specific contaminations during experiments', we wish to keep the current format.

Figure S3 – Repeated twice in the supplemental

We apologize for the mistake. The second Fig. 3 was deleted

Figure S4 – The x-axis on the right column states NH4 when I am assuming it should be NO3

We apologize for the mistake. The labels were revised from NH₄⁺ to NO₃⁻.

June 5, 2022

Dr. Takako Masuda
Japan Fisheries Research and Education Agency
Shiogama
Japan

Re: Spectrum02177-21R1 (*Crocospaera* as a major consumer of fixed nitrogen)

Dear Dr. Takako Masuda:

Your manuscript has been accepted, and I am forwarding it to the ASM Journals Department for publication. You will be notified when your proofs are ready to be viewed.

Sincerely,

Jeffrey Gralnick
Editor, Microbiology Spectrum
